# Identification of Novel Insulin Resistance Related ceRNA Network in T2DM and Its Potential Editing by CRISPR/Cas9

**DOI:** 10.3390/ijms22158129

**Published:** 2021-07-29

**Authors:** Marwa Matboli, Marwa Mostafa Kamel, Nada Essawy, Meram Mohamed Bekhit, Basant Abdulrahman, Ghada F. Mohamed, Sanaa Eissa

**Affiliations:** 1Medicinal Biochemistry and Molecular Biology Department, School of Medicine, Ain Shams University, Cairo 11566, Egypt; Marwa.kamel88@med.asu.edu.eg; 2Institut Pasteur, CEDEX 15, 75724 Paris, France; nada.essawy@pasteur.fr; 3Internal Medicine, Endocrinology and Diabetes Department, School of Medicine, Ain Shams University, Cairo 11566, Egypt; Merama_bekhet@yahoo.com; 4Calgary Prion Research Unit, University of Calgary, Calgary, AB T2N 4Z6, Canada; baabdulr@ucalgary.ca; 5Department of Biochemistry and Molecular Biology, Faculty of Pharmacy, Helwan University, Cairo 11795, Egypt; 6Department of Histology, School of Medicine, Ain Shams University, Cairo 11566, Egypt; ghadafaroukmohamed@gmail.com

**Keywords:** insulin resistance, lncRNA, miRNA, genes, gene editing, diabetes mellitus

## Abstract

Background: Type 2 diabetes mellitus is one of the leading causes of morbidity and mortality worldwide and is derived from an accumulation of genetic and epigenetic changes. In this study, we aimed to construct Insilco, a competing endogenous RNA (ceRNA) network linked to the pathogenesis of insulin resistance followed by its experimental validation in patients’, matched control and cell line samples, as well as to evaluate the efficacy of CRISPR/Cas9 as a potential therapeutic strategy to modulate the expression of this deregulated network. By applying bioinformatics tools through a two-step process, we identified and verified a ceRNA network panel of mRNAs, miRNAs and lncRNA related to insulin resistance, Then validated the expression in clinical samples (123 patients and 106 controls) and some of matched cell line samples using real time PCR. Next, two guide RNAs were designed to target the sequence flanking LncRNA/miRNAs interaction by CRISPER/Cas9 in cell culture. Gene editing tool efficacy was assessed by measuring the network downstream proteins *GLUT4* and *mTOR* via immunofluorescence. Results: *LncRNA-RP11-773H22.4*, together with *RET*, *IGF1R* and *mTOR mRNAs*, showed significant upregulation in T2DM compared with matched controls, while miRNA (i.e., *miR-3163 and miR-1*) and mRNA (i.e., *GLUT4* and *AKT2*) expression displayed marked downregulation in diabetic samples. CRISPR/Cas9 successfully knocked out *LncRNA-RP11-773H22.4*, as evidenced by the reversal of the gene expression of the identified network at RNA and protein levels to the normal expression pattern after gene editing. Conclusions: The present study provides the significance of this ceRNA based network and its related target genes panel both in the pathogenesis of insulin resistance and as a therapeutic target for gene editing in T2DM.

## 1. Introduction

Type 2 diabetes is a prevalent chronic metabolic disorder that represents a global growing healthcare burden, characterized mainly by resistance to insulin action along with insufficient secretion. The International Diabetes Federation (IDF) reported in 2019 that approximately 463 million adults (20–79 years old) are living with diabetes mellitus, and this number is expected to increase to 700 million by 2045 [1,2]. Egypt has topped the scale compared to other countries when it comes to the prevalence of DM. According to the IDF, Egypt’s diabetes prevalence (15.56% in adults of 20–79 years of age, with an annual mortality of 86,478 deaths) has placed it in the top 10 countries in terms of the number of diabetics [3].

Although T2DM is a multi-faceted disease with contributing genetic, epigenetic and environmental factors, reduced sensitivity to insulin in target cells constitutes its hallmark [4]. This factor represents the major pathological process underlying the disease that involves a variety of defects in signal transduction cascade, such defects in insulin Receptor Substrate 1/2 (IRS-1/2), phosphatidylinositol 3-Kinase (PI3K)/Serine/Threonine Kinase (*AKT*) and Glucose Transporter 4 (*GLUT4*) [5]. 

Epigenetics has been shown to be involved in the regulation of inflammation and cellular senescence, both of which are associated with type 2 diabetes. Studies postulate that levels of RNAs are affected by competition for a small pool of micro RNAs, a concept that has been described as the competing endogenous RNA theory (ceRNA), and is supported by growing evidence in a number of diseases [6]. Another type of non-coding RNA (ncRNA), long non-coding RNAs (lncRNAs), are implicated in diverse biological processes, especially epigenetic regulation. They can act as a ceRNA and influence post-transcriptional regulation. This is because they have miRNA binding sites and can affect the number of miRNAs levels available for binding to their target mRNAs. Consequently, they abolish the repression of these mRNAs, which explains their negative correlation with the miRNA’s level [7]. There is a list of long non-coding RNAs that have been linked to the development of T2DM, including HOTAIR, MEG3, LET and MALAT1 [8]. In addition to lncRNAs, many miRNAs are also implicated in the development of T2DM, namely, *miR-135, 202 and 214*, along with their targets (*Rock-1, Akt2,* and *Vamp2*) [9]. The potential cross-talk between miRNA–lncRNA–mRNA has been widely studied and ceRNA network analysis has proven to be an effective method of screening potential diagnostic and prognostic biomarkers in diverse diseases [10,11].

Gene editing is a recent breakthrough with a great potential for understanding and treating several diseases. Researchers could potentially target predetermined parts of the DNA sequence and produce specific alterations, including insertions, deletions, point mutations or translocations through the production of double-stranded breaks at target sites, activating DNA repair systems. Gene editing efficiency significantly improved after the discovery of a novel tool extracted from Streptococcus pyogenes, CRISPR (clustered regularly interspaced short palindromic repeats)-associated protein-9 nuclease (Cas9), which led to a potentially low cost, time saving method for editing the whole human genome (including non-coding RNAs) that is consistent with high throughput screening protocols [12].

The present paper aims to determine a potential insulin resistance related ceRNA panel from databases, and to validate its expression in both patients’ clinical samples and lymphocyte cell lines. Further, we call into question the role of this identified insulin resistance (IR) related panel in the pathogenesis of type 2 DM. Finally, we evaluate the efficacy of the gene editing tool CRISPR/Cas9 as a potential therapeutic strategy to modulate the expression of this deregulated network in T2DM. 

## 2. Results

### 2.1. Demographic, Clinical Data and Diabetes Laboratory Profile of the Diabetic and Healthy Groups

Demographic data regarding age, sex and smoking showed no significant difference (*p* > 0.05). There were significant differences in fasting and 2 h post prandial blood glucose level, BMI, blood pressure, lipid profile, albumin and creatinine ratio (alb./cr.), fasting insulin and glycosylated hemoglobin A1c (HbA1c), HOMA-IR between the diabetic and healthy groups (*p* < 0.05) (Table 1). 

### 2.2. Evaluation of the Identified ceRNA Panel Expression in the Diabetic and Healthy Groups

In T2DM, *LncRNA-RP11-773H22.4,* together with *RET*, *IGF1R* and *mTOR mRNAs* showed significant upregulation. On the other hand, the *miRNAs* (i.e., *miR-3163* and *miR-1*) and *mRNAs* (i.e., *GLUT4* and *AKT2*) displayed marked downregulation compared to the control group (Figure 1).

We calculated the sensitivity and specificity of each partner in the genetic network by ROC curve analysis (Appendix A). The overall positivity rates of these candidate genes in the diabetic and healthy groups are shown in Table 2.

### 2.3. Correlation between Different Partners in the Genetic Network, Glycemic Control and Insulin Resistance in Both Diabetic and Healthy Groups

There was a highly significant positive correlation between each of the following: *LncRNA-RP11-773H22.4, RET, IGF1R, mTOR mRNAs*, glycemic control and insulin resistance. On the other hand, we observed a marked negative correlation between each of the abovementioned parameters and *miR-3163, miR-1 miRNAs, GLUT4* and *AKT2 mRNAs* in both the diabetic and healthy groups (*p*< 0.01) using Spearman’s correlation, as shown in Table 3.

### 2.4. CRISPR/Cas9 Mediated Knockout of LncRNA-RP11-773H22.4 in Lymphocyte Cell Line Obtained from Diabetic Patients

#### 2.4.1. Effect of CRISPR/Cas9 Editing on Cell Count and Viability

By applying a one-way ANOVA post hoc test, we could not observe any difference in lymphocyte count and viability in the diabetic cell line before and after CRISPR/Cas9 editing. Of note, CRISPR editing slightly decreased the cell count and viability as compared to the healthy control, which may have been because of its potential genotoxic effect or the companion chemicals used for its transfection (Appendix A).

#### 2.4.2. Effect of CRISPR/Cas9 Editing on the Expression of the Identified Genetic Network

CRISPR/Cas9 gene editing modified the gene expression of the identified network, revealing the expression of *LncRNA-RP11-773H22.4*, *RET, IGF1R* and *mTOR mRNAs* to be significantly decreased (*p* < 0.01), and *miR-3163*, *miR-1*, *GLUT4* and *AKT2 mRNAs* to have obviously increased after CRISPR/Cas9 knockout of *LncRNA-RP11-773H22.4* (*p* < 0.01), as shown in Table 4.

There was a marked positive correlation between each of the following: *LncRNA-RP11-773H22.4, RET, IGF1R* and *mTOR mRNAs*. However, we noticed a highly significant negative correlation between each of the above-mentioned parameters and *miR-3163, miR-1 miRNAs, GLUT4* and *AKT2 mRNAs* among the diabetic groups before and after CRISPR/Cas9 editing, as well as with the control group (*p* < 0.01) using Spearman’s correlation (Table 5).

#### 2.4.3. Effect of CRISPR/Cas9 Editing on *GLUT4* and *mTOR* Proteins 

Lymphocytes obtained from the healthy donor “normal pool” showed compact lymphocytes colonies with *GLUT4* high fluorescence intensity and loss of *mTOR* expression (Figure 2a,b). On the other hand, lymphocytes obtained from T2DM showed a merged collection “small colony” of cells that were uniformly distributed with an enlarged size, high nuclear/cytoplasmic ratio and irregular nuclear contour, with membrane vesicles and marked *mTOR* fluorescence intensity (+++) in the cytoplasm and nucleus along with faint *GLUT4* expression (+). After CRISPR editing, discrete spread-out lymphocytes of medium cell sizes were uniformly distributed, with slightly regular nuclei, moderate nuclear/cytoplasmic ratios and few membranous vesiculations that had moderate faint fluorescence of *mTOR* (+) and dense clustering expressions of *GLUT4* with a high fluorescence intensity (+++).

## 3. Discussion

Type 2 diabetes is the most common form of diabetes, accounting for almost 90% of all cases of this debilitating illness. Characteristically, T2DM is defined by both insulin resistance and pancreatic beta cell malfunction, leading to hyperglycemia [13].

According to the screened insulin resistance candidate genes, their interacting miRNAs and associated lncRNAs, a ceRNA network was constructed based on their biological interactions. Although each step might involve a large number of candidate mRNAs, miRNAs or lncRNAs, only several of them were used to construct the ceRNA network after a step-by-step screening and validation process. The screened hub genes were differently expressed, and our results demonstrated that there was significant upregulation in the expression of *LncRNA-RP11-773H22.4, RET, mTOR* and *IGF1R mRNAs,* together with significant downregulation in the expression of *miR-3163, miR-1, GLUT4* and *AKT2 mRNAs* in the diabetic group compared to the healthy control in both patients’ clinical samples and lymphocyte cell lines.

We studied the influence of each candidate gene in the insulin signaling pathway at various stages so as to predict the pattern of dysregulation of these genes and how they modulate other epigenetic effectors in insulin resistance. Regarding the *RET* gene, it is a protooncogene residing on chromosome 10q11.2 coding for a tyrosine kinase receptor that has never been reported in T2DM. It participates in many intracellular signaling pathways, including the *PI3K–AKT* and *MAPK–ERK*. The involvement of *RET* oncogene in the same pathway of insulin signaling explains its role in the molecular pathogenesis of insulin resistance [14,15]. On the other hand, *glucose transporter 4* (*GLUT4*), which is coded by the *SLC2A4* gene, is the center of most studies on insulin resistance, since it is the major means of glucose transport in insulin-sensitive peripheral tissues (i.e., skeletal muscles and adipose tissues) [16,17]. Another important mediator in the PI3K pathway involved in insulin signaling is *Akt2*, which influences numerous downstream proteins that affect metabolism, growth and cell survival. *Akt2* expression is highest in insulin-sensitive tissues and is believed to contribute largely to the role of insulin in metabolism [18]. Therefore, derangement in its protein product may contribute to the pathogenesis of insulin resistance and/or T2DM. Multiple studies have specified that *Akt2* is responsible for insulin-dependent glucose uptake in humans as well as rodents, and its dysfunction is related to insulin resistance and impaired glucose tolerance, which goes hand in hand with our study [19].

One of our screened candidate genes, *insulin-like growth factor-1 receptor (IGF1R)*, is a tyrosine kinase receptor that is critical for insulin signaling. *IGF1R* is now considered a key player in the activation of the phosphatidylinositol 3-kinase–AKT and to be one of the important receptor proteins for the insulin signaling pathway. Specifically, *IGF1* and its receptor affect the sensitivity of muscle to insulin [20,21].

Another important key player in the phosphatidylinositol 3-kinase (PI3K)-related kinase family and a member in our constructed network is mTOR, a special type of multi-subunit serine/threonine kinase. Surprisingly, *mTOR* is the target of the valuable drug rapamycin, which is used to coat coronary stents and prevent organ transplant rejection. This versatility can be explained by the critical role of *mTOR* in response to miscellaneous signaling cascades when stimulated by various intracellular and environmental conditions [22]. Being a major node in the (*PI3K/AKT/mTOR*) pathway, we think that its disruption will result in hyperglycemia and diabetes [23].

By combining microRNAs with T2DM-related target genes, we identified microRNA–lncRNA interaction pairs associated with T2DM. These small ncRNAs are important negative regulators in coding noncoding RNA networks. The miRNA:mRNA interactions were validated based on their biological and expression relationships. The role of miRNAs has been delineated in several studies related to various critical protein cascades that share in insulin signaling pathways [24]. Among the most notable of these miRNAs is *micro-RNA-1*, described in several diseases and cancers. *micro-RNA-1* is highly expressed in muscle cells where it suppresses the proliferation of precursor cells and encourages myogenesis [25]. *miR-1* downregulation has been shown to be related to diabetes-induced oxidative stress. A study by Chen et al. revealed that under oxidative stress insulin regulates the function of *miR-1* through *AKT* activation in H9c2 cells [26]. These findings confirm our study results, which revealed deregulation of *miR-1* levels in T2DM [27].

In the current study, a newly investigated miRNA (i.e., *miR-3163*) in T2DM was retrieved from databases based on its association with many downstream effectors in the insulin signaling pathway [28,29]. We found that its levels were significantly deregulated in T2DM in both clinical and cell line samples.

Interestingly, Chakraborty et al. found that *miR-1* influences the expression of both IGF-1 and its receptor. Through modulating *IGF1* and *IGF1R, miR-1* promotes insulin resistance in the endothelium and adipocytes [30], which may explain the relationship between them in our established T2DM related network

Recent studies have reported that lncRNAs may work as competing endogenous RNAs (ceRNAs) and may crosstalk with mRNAs via competitive binding with their common pools of miRNAs in most human diseases [31].

Remarkably, our study demonstrated that *Lnc-RNA-RP11-773H22.4* is expressed in patients with T2DM, and it was correlated with poor glycemic control and insulin resistance. These non-coding RNAs have been proven to be involved in many nodes in the development of cancer and DM by affecting the expression of the disease associated genes at the epigenetic level, as well as transcriptional and posttranscriptional levels [8,32].

We observed a highly significant positive correlation between the expressions of *LncRNA-RP11-773H22.4, RET, mTOR mRNAs*, HbA1c and HOMA-IR. Meanwhile, a markedly negative correlation was found between them and *miR-3163* and *miR-1 miRNA* expression in the diabetic and healthy groups (*p* < 0.01). These correlations have been validated in both clinical samples and in lymphocyte cell line, which agrees with the in-silico data analysis. We believe that *LncRNA-RP11-773H22.4* has an inhibitory effect on *miR-3163 miRNA,* and thus is responsible for the inhibitory effect of that miRNA on *RET mRNA* and other mRNAs in the panel.

Importantly, our study provides evidence that the CRISPR/Cas9 genome editing system is a simple, cheap and fast tool by which to manipulate genomes that are expected to have a broader therapeutic application in insulin resistance by modulating the deregulated genetic network of the insulin signaling pathway (Figure 3).

Taken together, we applied for the first time an approach that combined a computational method with clinical validation, to provide novel insights into the molecular mechanisms of insulin resistance. We implemented a combined bioinformatics analysis to retrieve a set of ceRNA networks (*LncRNA-RP11-773H22.4, miR-1, miRNA-3163,* which is related to insulin resistance and their targeting signaling pathway genes *RET, IGF1R, GLUT4, AKT2* and *mTOR mRNAs*) from public databases. Afterwards, we investigated the identified ceRNA network expression in patients’ clinical samples and lymphocyte cell lines via qPCR and IF. We knocked out the *LncRNA-RP11-773H22.4* from the sequence of lncRNA–miRNA interaction in the cultured lymphocytes using CRISPR/Cas9. This approach resulted in the restoration of normal expression of the identified genes in comparison to the healthy controls, as confirmed by qPCR and IF. We therefor suggest the CRISPR/Cas9 knockout of *LncRNA-RP11-773H22.4* as a potential therapeutic target for treatment of T2DM.

More in vitro functional studies are needed to validate our results and to consider the genotoxicity and off target mutation safety issues related to the CRISPR/Cas9 system, with the possibility of using more accurate nucleases to reduce off-target effects.

## 4. Material and Methods

### 4.1. Participants and Samples

Participants were divided into two groups, the first comprising 123 cases with type 2 diabetes mellitus. T2DM diagnosis was proved according to the American Diabetes Association practice guidelines. Ages ranged from 35 to 83 years, with a mean of 54.69 ± 8.855 years and a median age of 55 years. The second group comprised 106 healthy normal individuals undergoing a routine checkup at Ain Shams University hospitals. Their ages ranged from 36 to 69 years, with a mean age 53.29 ± 7.04 years and a median age of 53 years. Age and sex were matched between the two groups.

Inclusion criteria of the stud were a proven diagnosis of T2DM according to ADA practice guidelines, an age > 18 years at the time of consent giving and the ability to provide a written, informed consent. Exclusion criteria were a history of malignant disease, use on steroids for the last 6 months, end stage organ disease as chronic liver disease and pregnancy or lactation.

Whole blood samples were collected at Ain Shams University hospitals during the March 2018 to May 2019 period. Sera were obtained by centrifugation, while peripheral blood mononuclear cells (PBMCs) were isolated using lymphoprep (Axis-Shield PoC AS, Oslo, Norway). All participants signed an informed consent, and the study was approved by the ethical committee of the Faculty of Medicine, Ain Shams University, Egypt, dated 19/4/2017, FWA 000017585.4.2. Measurement of HOMA-IR

Fasting insulin level was measured in sera of diabetic patients and healthy controls using enzyme-linked immunosorbent assay ELISA (DRG Insulin ELISA (EIA-2935), DRG International, Inc., Springfield, NJ, USA). A Homeostatic Model Assessment of Insulin Resistance (HOMA-IR) was calculated according to the equation: fasting insulin (μU/L) × fasting glucose (nmol/L)/22.5 [33].

### 4.2. Bioinformatics-Based Selection of Insulin Resistance Related ceRNA Panel

The identified panel was obtained through the following steps:

Retrieval of a set of candidate genes (mRNAs) related to insulin resistance signaling pathways from two public microarray databases available at (https://www.ebi.ac.uk/gxa/home, accessed on 9 June 2017) and (https://www.proteinatlas.org, accessed on 9 June 2017). Verification of the identified candidate gene expression in skeletal muscle and adipose tissues through gene cards database (https://www.genecards.org/, accessed on 9 June 2017), so as to decrease the false discovery rate. Construction of mRNA–miRNA–lncRNA genetic axis linked to the identified candidate genes in IR in T2DM including *LncRNA-RP11-773H22.4*, *miR-3163* and *miR-1 microRNAs* using a lnCeDB database, available at (http://gyanxet-beta.com/lncedb/index.php, accessed on 9 June 2017). Alignment between *LncRNA-RP11-773H22.4* and retrieved miRNAs (*miR-3163* and *miR-1*)*;* alignment between *mTOR mRNA* and retrieved miRNAs; and alignment between *LncRNA-RP11-773H22.4* and the two synthesized gRNAs was used to identify sequences of lncRNA–miRNA interactions to be targeted by the CRISPR/Cas9 gene editing tool and to identify sequences of coding and, non-coding gene interactions through the alignment database tool available at (https://www.ebi.ac.uk/gxa/home, accessed on 9 June 2017). Detailed bioinformatics are shown in Appendix A.

### 4.3. Validation of the Identified Insulin Resistance Related ceRNA Panel in Human Clinical Samples

#### 4.3.1. Extraction of Total RNA, Including Coding and Non-Coding RNAs from Human PBMCs Samples

We used the miRNeasy RNA isolation kit (Cat no. 74104; Qiagen, Valencia, CA, USA) to extract total RNA from the PBMCs following the manufacturer’s instructions. We assessed the RNA concentration and integrity using a DeNovix DS-11 micro-volume spectrophotometer (Wilmington, NC, USA). Then, total extracted RNA were reverse transcribed into cDNA using the RT2 first strand kit (Cat no. 330401; Qiagen, Valencia, CA, USA) for the target mRNAs, and miScript II RT Kit (Cat no. 218161; Qiagen, Valencia, CA, USA) for the non-coding RNAs, as per the manufacturer’s protocol using Thermo Hybaid PCR express (Thermo Scientific, Waltham, MA, USA).

#### 4.3.2. Quantitative Real Time-PCR of the Identified Insulin Resistance Related ceRNA Panel

The levels of the identified *mRNAs (RET, IGF1R, m-TOR, GLUT4* and *AKT2)* in PBMCs were measured using a custom RT^2^ Profiler PCR Array (Cat no. 330171; Qiagen, Helman Germany, Ensembl: ENSG00000165731, ENSG00000140443, ENSG00000198793, ENSG00000181856 and ENSG00000105221, respectively) and RT² SYBR Green ROX qPCR Mastermix (Cat no: 330520; Qiagen, Helman, Germany) with an Applied Biosystems 7500 Real-Time PCR system (Foster city, CA, USA). Relative expression levels for *LncRNA-RP11-773H22.4* were analyzed via the RT² SYBR Green ROX qPCR Master mix (Cat no: 330520; Qiagen, Helman, Germany) and lncRNA qPCR Primer Assay for Human RP11-773H22.4 (ENST00000588211), supplied by Qiagen. *GAPDH* (Ensembl: ENSG00000111640) was used as a reference gene. *miR-3163* and *miR-1 miRNAs* relative expression levels in PBMCs were investigated using a miScript SYBR Green PCR Kit (Cat no. 218073; Qiagen, Helman, Germany), a miScript universal primer and a miRNA-specific forward primer (Hs_miR-3163_1 miScript Primer Assay) (Accession: MIMAT0015037) (Cat no. MS00020769; Qiagen, Helman, Germany) for *miR-3163* and (Hs_mir-1-1_PR_1 miScript Precursor Assay) (Accession: MI0000651) (Cat no. MP00000175; Qiagen, Helman, Germany) *miR-1*; *RNU-6* was used as an internal control. All PCR primers were obtained from Qiagen. The PCR program cycling conditions were adjusted according to the type of RNA measured according to the manufacturer’s protocol. The 2^−ΔΔCt^ technique was used to measure the expression of the IR-specific RNA-based candidate genes panel using Applied Biosystems 7500 software v2.3. Reference genes were used as an internal control to normalize the raw data of the samples and compare these results to a reference sample. In this study, appropriate standardization strategies were carried out to recognize any experimental error introduced at any stage during extraction and processing of the RNA according to MIQE guidelines [34].

### 4.4. Validation of the Identified Insulin Resistance related ceRNA Panel and CRISPR/Cas9 Editing of LncRNA-RP11-773H22.4 in Lymphocyte Cell Line

#### 4.4.1. Culture of Human Lymphocytes

Human cell culture was conducted as previously described [35]. The PBMCs were transferred in 20 mL RPMI 1640 media (cat. no. 11875093) to a T-75 culture flask containing 10% fetal bovine serum, 1% penicillin/streptomycin and 1 μg/mL phytohemagglutinin (PHA), which were incubated at 37 °C with 5% CO_2_ for 24 h. The next day, all media were removed from the flask, and the cell pellet was added to a 50 mL conical tube and centrifuged at 500× *g* for 5 min. The pellet that contained mainly lymphocytes was resuspended. The cells were transferred to a new T-75 culture flask in 25 mL RPMI 1640 media containing 10% fetal bovine serum, 1% penicillin/streptomycin and 1 μg/mL PHA and incubated at 37 °C for 3 days. After 24 h of growth, 20 mL of fresh media was added and transferred to a larger T-175 culture flask. After 3 days, media and suspended cells were removed from the culture flask and transferred to a 50 mL conical tube, then centrifuged at 500× *g* for 5 min. The pellet was resuspended and transferred to a new T-75 culture flask containing 25 mL RPMI 1640 with 10% fetal bovine serum and 1% penicillin/streptomycin. Lymphocytes were grown for 4 days.

#### 4.4.2. Synthesis of gRNAs

DNA oligonucleotides used for gRNA synthesis were designed with the GeneArt CRISPR gRNA Design Tool available at (www.thermofisher.com, accessed on 9 June 2017). The two gRNAs were then synthesized using the GeneArt Precision gRNA Synthesis Kit (Cat no. A29377, Thermofischer, Waltham, MA, USA) according to the manufacture’s protocol. We estimated the concentration of gRNA by using the Qubit RNA BR Assay Kit (cat. no. Q10210). Each gRNA was combined with GeneArt Platinum Cas9 Nuclease (cat. no. B25640, Thermofischer, Waltham, MA, USA) to form the Cas9 protein/gRNA ribonucleoprotein complexes (Cas9 RNPs), as seen in Appendix A and Appendix A.

#### 4.4.3. Lymphocyte Transfection in a 12-Well Plate Using Lipofectamine CRISPRMAX Reagent (cat. no. CMAX00008, Thermofischer, Waltham, MA, USA)

At the transfection day, 1.5 mL sterile Eppendorf tube was filled with 50 μL of Opti-MEM medium (cat. no. 31985062), then 1.25 µg GeneArt Platinum Cas9 nuclease and 0.25 µg gRNA were added followed by mixing using vortex, after that, 2.5 μL Cas9 Plus reagent was added to the mixture that contained Cas9 protein and gRNA, mixed well, and incubated at 25 °C for 5 min to form RNPs. Meantime, 3 µL of Lipofectamine CRISPRMAX was mixed with 50 µL Opti-MEM and incubated at 25 °C for 5 min before being added to the RNP solution. The mixture was incubated at 25 °C for 15 min, then added to cells that were plated onto 12-well plates at a density of 8.5 × 10^5^ cells/well in 1 mL growth medium. At 72 h post-transfection, lymphocytes were harvested for cell count and viability using the trypan blue exclusion method, and gene expression was analyzed before and after editing [36]. The genomic cleavage efficiency was measured by the GeneArt Genomic Cleavage Detection kit (cat. no. A24372, Thermofischer, Waltham, MA, USA).

#### 4.4.4. Immunofluorescence Microscopy to Evaluate the Activity of *GLUT4* and *mTOR* Proteins

The harvested cells were examined by immunofluorescence staining with specific polyclonal antibodies against *solute carrier family 2 member4 (SLC2A4/GLUT4)* (cat. no. FNab03503, FineTest, Wuhan, China) and mammalian target of rapamycin (m-TOR) protein (cat. no. FNab05417, FineTest, Wuhan, China) (green) at dilutions 1:100. An Alexa Fluor 488 anti-rabbit IgG secondary antibody (cat. no. A-11034, Invitrogen; ThermoFisher Scientific, Hilden, Germany) was used for detection at dilution 1:100 in antibody dilution buffer. Fluorescence was examined by an immunofluorescence microscope LX400 (cat. no. 9126000, Labmed; USA), using Optika ISview image acquisition and processing software.

### 4.5. Statistics

Statistical analyses were carried out using Statistical Package for the Social Sciences (SPSS, Chicago, IL, USA) software version 20. Qualitative data were described using the number and percentage. Quantitative data were described using mean rank for non-parametric data and mean, standard deviation for parametric data. The significance of the obtained results was judged at the (0.05) level. Qualitative data were compared using a Chi-square test. Quantitative data were compared using a Student’s *t*-test and one-way ANOVA test; a Kruskal–Wallis test and Mann–Whitney U test was used to compare independent groups (for non-parametric data). ROC curves were used to explore the predictive value of investigated panel. The relationships between the investigated parameters were assessed using Spearman rank correlations. A two-tailed *p* value of ≤0.05 was considered statistically significant.

### 4.6. Data and Resource Availability

The datasets and RESOURCE generated during and/or analyzed during the current study are available from the corresponding author upon reasonable request.

## Figures and Tables

**Figure 1 ijms-22-08129-f001:**
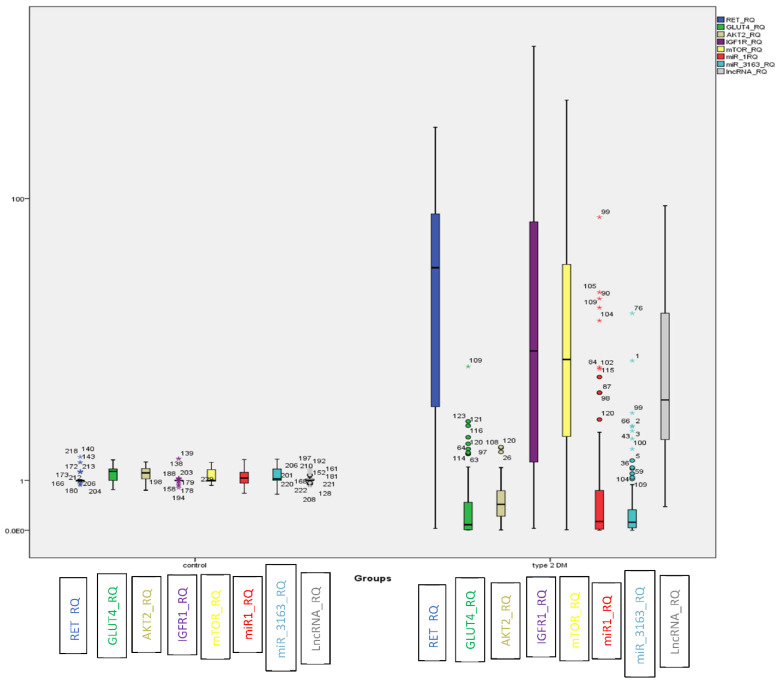
BOXPLOT showing differential expression of non-coding genes *LncRNA-RP11-773H22.4, miR-3163 and miR-1 miRNAs* and candidate genes *RET, IGF1R, m-TOR, GLUT4* and *AKT2* as determined by qRT-PCR in the investigated groups ((o) identify outliers and (*) identify extreme outliers).

**Figure 2 ijms-22-08129-f002:**
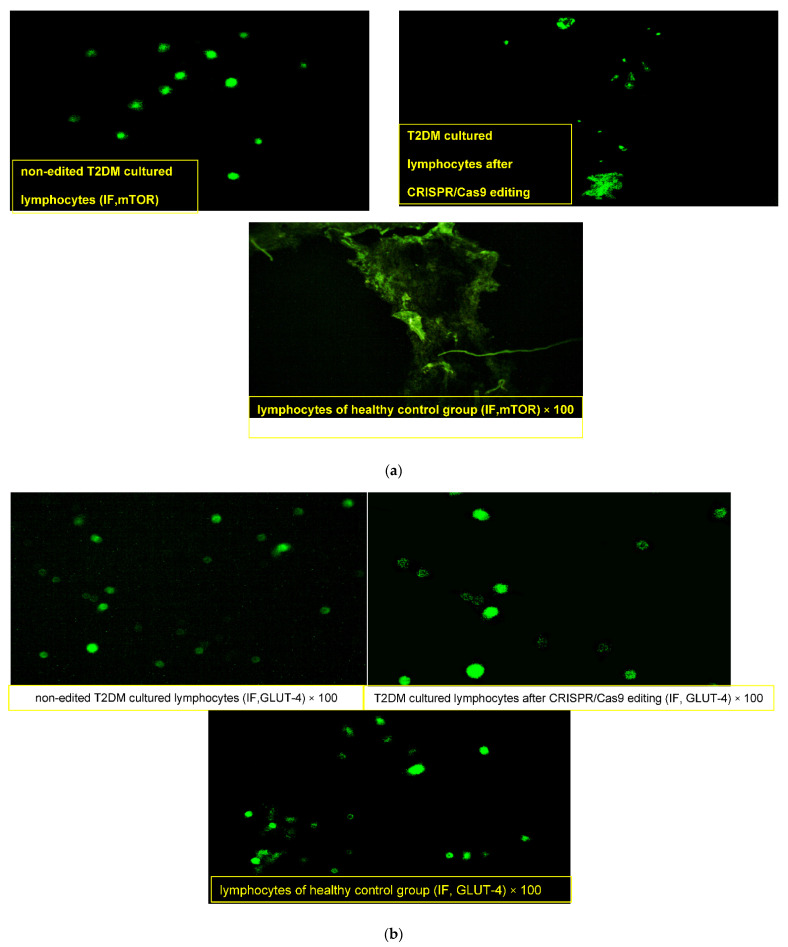
(**a**) Immunofluorescence assay staining of lymphocyte cultured cells with specific polyclonal antibodies against mammalian target of rapamycin (m-TOR) protein. (**b**) Immunofluorescence assay staining of lymphocyte cultured cells with specific polyclonal antibodies against solute carrier family 2 (facilitated glucose transporter), member 4 (GLUT4) protein.

**Figure 3 ijms-22-08129-f003:**
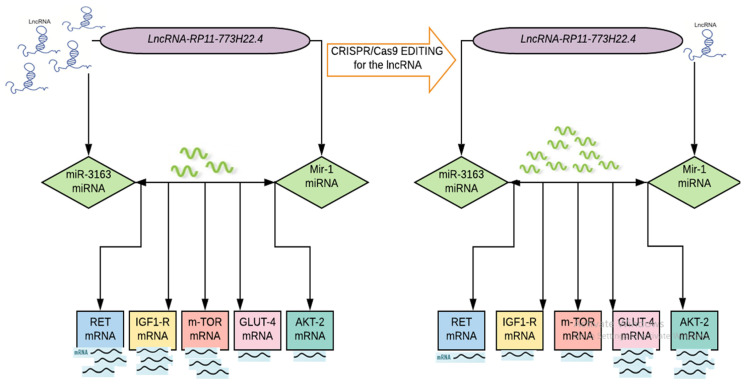
Proof of concept of ceRNA hypothesis in IR in T2DM.

**Table 1 ijms-22-08129-t001:** Study demographic data between the diabetic and healthy groups (*n* = 229).

	Type 2 DM N (%)	Healthy N (%)	χ^2^(P) ^(a)^
Sex: Male (87) Female (142)	47 (38.2%) 76 (61.8%)	40 (37.7%) 66 (62.3%)	0.005 (0.941)
Smoking: smoker: (102) non-smoker:(117) x-smoker: (10)	86 (69.9%) 30 (24.4%) 7 (5.7%)	60 (56.7%) 46 (43.3%) 3 (2.8%)	0.004 (0.870)
Family history of diabetes: Positive: (98) Negative: (131)	98 (79.7%) 25 (20.3%)	0 (0%) 106 (100%)	147.636 (0.000 **)
Oral Anti Diabetic Medications (OAD): Metformin: (44) SU: (24) Combined: (12) Not Taking OAD: (149)	44 (35.8%) 24 (19.5%) 12 (9.8%) 43 (35%)	NA	NA
	**Mean ± SD**	**Mean ± SD**	**t(P) ^(b)^**
Duration of diabetes (yrs)	9.38 ± 6.067	NA	NA
Fasting blood glucose (mg/dL)	197 ± 91.852	88.08 ± 14.594	12.173 (0.000 **)
2hpp glucose (mg/dL)	274.79 ± 99.6	111.07 ± 16.926	16.712 (0.000 **)
Systolic Bl.pr. (mmHg)	135.45 ± 15.59	118.02 ± 7.736	10.452 (0.000 **)
Diastolic Bl.pr. (mmHg)	88.09 ± 11.969	76.56 ± 5.088	9.226 (0.000 **)
BMI (kg/m^2^)	34.05 ± 5.71	30.67 ± 4.93	2.347 (0.000 **)
Total Cholesterol (mg/dL)	292.46 ± 60.866	102.97 ± 24.47	30.023 (0.000 **)
TGs (mg/dL)	263.84 ± 75.914	90.16 ± 19.62	22.766 (0.000 **)
HDL (mg/dL)	31.98 ± 9.6	71.18 ± 11.235	−28.458 (0.000 **)
LDL (mg/dL)	200.77 ± 55.287	75.49 ± 17.861	22.341 (0.000 **)
alb./cr. (mg/mg)	24.39 ± 4.625	13.59 ± 3.349	19.944 (0.000 **)
Hb A1c (gm%)	7.91 ± 2.11	3.89 ± 1.205	17.307 (0.000 **)
	**Mean Rank**	**Mean Rank**	**U(P) ^(c)^**
Fasting insulin (uIU/mL)	140.52	85.38	3379.5 (0.000 **)
HOMA-IR	161.67	60.84	778 (0.000 **)

^(a)^ Chi-square test, ^(b)^ independent samples *t* test, ^(c)^ Mann–Whitney test, ** *p* < 0.01: is highly significant, FBG: fasting blood glucose, 2hPP BG: 2 h postprandial blood glucose, BMI: body mass Index, HDL: high density lipoprotein, LDL: low density lipoprotein, TGs: triglycerides, alb./cr.: albumin/creatinine ratio, HbA1c: hemoglobin A1c, HOMA-IR: Homeostatic Model Assessment of Insulin Resistance.

**Table 2 ijms-22-08129-t002:** Positivity rate of HOMA-IR and the investigated candidate genes in the investigated groups according to the calculated cutoff (*n* = 229).

	Type 2DM N (%)	Healthy N (%)	χ^2^(P) ^(a)^
*HOMA-IR* Positive ≥ 2.3 (102) Negative < 2.3 (127)	102 (82.9%) 21 (17.1%)	0 (0%) 106 (100%)	158.501 (0.000 **)
*LncRNA-RP11-773H22.4 RQ:* Positive ≥ 1.2 (124) Negative < 1.2 (105)	117 (95.1%) 6 (4.9%)	7 (6.6%) 99 (93.4%)	179.680 (0.000 **)
*miR-3163 RQ:* Positive ≤ 0.91 (120) Negative > 0.91 (109)	109 (88.6%) 14 (11.4%)	11 (10.4%) 95 (89.6%)	139.734 (0.000 **)
*miR-1 RQ:* Positive ≤0.87 (116) Negative > 0.87(113)	96 (78%) 27 (22%)	20 (18.9%) 86 (81.1%)	79.776 (0.000 **)
*RET mRNA RQ:* Positive ≥ 1.36 (117) Negative < 1.36 (112)	114 (92.7%) 9 (7.3%)	3 (2.8%) 103 (97.2%)	183.952 (0.000 **)
*mTOR mRNA RQ:* Positive ≥ 1.4 (116) Negative < 1.4 (113)	105 (85.4%) 18 (14.6%)	11 (10.4%) 95 (89.6%)	128.085 (0.000 **)
*IGF1R mRNA RQ:* Positive ≥ 1.07 (100) Negative < 1.07 (129)	98 (79.7%) 25 (20.3%)	2 (1.9%) 104 (98.1%)	140.05 (0.000 **)
*GLUT4 mRNA RQ:* Positive ≤ 0.92 (116) Negative > 0.92 (113)	109 (88.6%) 14 (11.4%)	7 (6.6%) 99 (93.4%)	153.210 (0.000 **)
*AKT2 mRNA RQ:* Positive ≤ 0.99 (118) Negative > 0.99 (111)	113(91.9%) 10 (8.1%)	5 (4.7%) 101 (95.3%)	173.143 (0.000 **)

^(a)^ Chi-square test, p: NS, not significant (>0.05), ** *p* < 0.01: is highly significant.

**Table 3 ijms-22-08129-t003:** Correlation between the investigated parameters expression in both the diabetic and healthy groups (*n* = 229).

		*lncRNA_RQ*	*miR_3163_RQ*	*miR_1RQ*	*RET_RQ*	*IGF1R_RQ*	*mTOR_RQ*	*GLUT4_RQ*	*AKT2_RQ*	HbA1c	HOMA_IR
*lncRNA_RQ*	Correlation Coefficient (Sig.)	1.000	−0.515 (0.000)	−0.366 (0.000)	0.656 (0.000)	0.483 (0.000)	0.469 (0.000)	−0.579 (0.000)	−0.588 (0.000)	0.651 (0.000)	0.581 (0.000)
*miR_3163_RQ*	Correlation Coefficient (Sig.)	−0.515 (0.000)	1.000	0.459 (0.000)	−0.475 (0.000)	−0.330 (0.000)	−0.399 (0.000)	0.455 (0.000)	0.505 (0.000)	−0.576 (0.000)	−0.555 (0.000)
*miR_1RQ*	Correlation Coefficient (Sig.)	−0.366 (0.000)	0.459 (0.000)	1.000	−0.457 (0.000)	−0.290 (0.000)	−0.259 (0.000)	0.377 (0.000)	0.412 (0.000)	−0.362 (0.000)	−0.378 (0.000)
*RET_RQ*	Correlation Coefficient (Sig.)	0.656 (0.000)	−0.475 (0.000)	−0.457 (0.000)	1.000	0.694 (0.000)	0.440 (0.000)	−0.491 (0.000)	−0.615 (0.000)	0.562 (0.000)	0.543 (0.000)
*IGF1R_RQ*	Correlation Coefficient (Sig.)	0.483 (0.000)	−0.330 (0.000)	−0.290 (0.000)	0.694 (0.000)	1.000	0.392 (0.000)	−0.362 (0.000)	−0.420 (0.000)	0.462 (0.000)	0.296 (0.000)
*mTOR_RQ*	Correlation Coefficient (Sig.)	0.469 (0.000)	−0.399 (0.000)	−0.259 (0.000)	0.440 (0.000)	0.392 (0.000)	1.000	−0.421 (0.000)	−0.488 (0.000)	0.547 (0.000)	0.415 (0.000)
*GLUT4_RQ*	Correlation Coefficient (Sig.)	−0.579 (0.000)	0.455 (0.000)	0.377 (0.000)	−0.491 (0.000)	−0.362 (0.000)	−0.421 (0.000)	1.000	0.523 (0.000)	−0.517 (0.000)	−0.573 (0.000)
*AKT2_RQ*	Correlation Coefficient (Sig.)	−0.588 (0.000)	0.505 (0.000)	0.412 (0.000)	−0.615 (0.000)	−0.420 (0.000)	−0.488 (0.000)	0.523 (0.000)	1.000	−0.614 (0.000)	−0.608 (0.000)
HbA1c	Correlation Coefficient (Sig.)	0.651 (0.000)	−0.576 (0.000)	−0.362 (0.000)	0.562 (0.000)	0.462 (0.000)	0.547 (0.000)	−0.517 (0.000)	−0.614 (0.000)	1.000	0.611 (0.000)
HOMA_IR	Correlation Coefficient (Sig.)	0.581 (0.000)	−0.555 (0.000)	−0.378 (0.000)	0.543 (0.000)	0.296 (0.000)	0.415 (0.000)	−0.573 (0.000)	−0.608 (0.000)	0.611 (0.000)	1.000

Spearman’s correlation: NS, not significant (>0.05), *p* < 0.01: is highly significant, *p* < 0.05: is significant. HbA1c: hemoglobin A1c, HOMA-IR: Homeostatic Model Assessment of Insulin Resistance.

**Table 4 ijms-22-08129-t004:** Differential expression of *LncRNA***-***RP11-773H22.4, miR-3163 and miR-1 miRNAs RET, IGF1R, m-TOR, GLUT4* and *AKT2 mRNAs* between lymphocyte cell lines before and after CRISPR/Cas9 editing.

	Lymphocytes’ Cell Lines		U(P) ^(a)^
	T2DM Lymphocytes without Editing	T2DM Lymphocytes with CRISPR Editing	
	Mean Rank	Mean Rank	
*LncRNA-RP11-773H22.4* RQ	14	5	0.000 (0.000 **)
*miR-3163 miRNA RQ*	5	14	0.000 (0.001 **)
*miR-1 miRNA RQ*	5	14	0.000 (0.001 **)
*RET mRNA RQ*	*14*	*5*	0.000 (0.000 **)
*IGF1R mRNA RQ*	*14*	*5*	0.000 (0.000 **)
*mTOR mRNA RQ*	*14*	*5*	0.000 (0.000 **)
*GLUT-4 mRNA RQ*	*5*	14	0.000 (0.000 **)
*AKT2 mRNA RQ*	*5*	14	0.000 (0.000 **)

**^(^**^a)^ Mann–Whitney test ** *p* < 0.01: is highly significant. RQ: relative quantity. Healthy normal RQ values are not included as they are the control group (values around 1) and the above RQ values are calculated from it.

**Table 5 ijms-22-08129-t005:** Correlation between the investigated candidate genes expression in the three investigated lymphocyte cell lines (*n* = 27).

		*lncRNA_RQ*	*miR_3163_RQ*	*miR_1 RQ*	*RET_RQ*	*IGF1R_RQ*	*mTOR_RQ*	*GLUT4RQ*	*AKT2_RQ*
*lncRNA_RQ*	Correlation Coefficient (Sig.)	1.000	−0.931 (0.000)	−0.983 (0.000)	0.914 (0.000)	0.931 (0.000)	0.879 (0.000)	−0.897 (0.000)	−0.879 (0.000)
*miR_3163_RQ*	Correlation Coefficient (Sig.)	−0.931 (0.000)	1.000	0.965 (0.000)	−0.948 (0.000)	−0.965 (0.000)	−0.931 (0.000)	0.913 (0.000)	0.948 (0.000)
*miR_1RQ*	Correlation Coefficient (Sig.)	−0.983 (0.000)	0.965 (0.000)	1.000	−0.931 (0.000	−0.948 (0.000)	−0.897 (0.000)	0.879 (0.000)	0.897 (0.000)
*RET_RQ*	Correlation Coefficient (Sig.)	0.914 (0.000)	−0.948 (0.000)	−0.931 (0.000)	1.000	0.983 (0.000)	0.983 (0.000)	−0.919 (0.000)	−0.931 (0.000)
*IGF1R_RQ*	Correlation Coefficient (Sig.)	0.931 (0.000)	−0.965 (0.000)	−0.948 (0.000)	0.983 (0.000)	1.000	0.948 (0.000)	−0.931 (0.000)	−0.948 (0.000)
*mTOR_RQ*	Correlation Coefficient (Sig.)	0.879 (0.000)	−0.931 (0.000)	−0.897 (0.000)	0.983 (0.000)	0.948 (0.000)	1.000	−0.931 (0.000)	−0.948 (0.000)
*GLUT4RQ*	Correlation Coefficient (Sig.)	−0.897 (0.000)	0.913 (0.000)	0.879 (0.000)	−0.914 (0.000)	−0.931 (0.000)	−0.931 (0.000)	1.000	0.983 (0.000)
*AKT2_RQ*	Correlation Coefficient (Sig.)	−0.879 (0.000)	0.948 (0.000)	0.897 (0.000)	−0.931 (0.000)	−0.948 (0.000)	−0.948 (0.000)	0.983 (0.000)	1.000

Spearman’s correlation: NS, not significant (>0.05), *p* < 0.01: is highly significant, *p* < 0.05: is significant.

## Data Availability

The datasets generated during and/or analyzed during the current study are available from the corresponding author on reasonable request.

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
