# Peer review of "Identification of Novel Insulin Resistance Related ceRNA Network in T2DM and Its Potential Editing by CRISPR/Cas9"

_ijms, 2021, doi:10.3390/ijms22158129_

Round 1
Reviewer 1 Report
The manuscript entitled „Identification of novel insulin resistance related ceRNA network: LncRNA, RP11‐773H22.4; miR-3163, miR-1; mRNAs:RET , IGF1-R, m-TOR, GLUT-4, AKT2 in T2DM and its potential editing by CRISPR/Cas9” describe a innovative research aimed at constructing in siilco, a competing endogenous RNAs (ceRNAs) network linked to the pathogenesis of insulin resistance followed by its experimental validation in patients evaluation the efficacy of CRISPR/Cas9 as a potential therapeutic strategy to modulate the expression of this deregulated network in type 2 diabetic patients. The research was designed appropriately. The authors also describe all its phases in a very detailed way. Moreover the manuscript is very interesting and well written. Therefore, I have no major comments on its current form. However, please follow the manuscript from an editorial perspective as there are numerous spelling errors like eg.
- „invitro” without space.
- two commas between authors
- please check also gene and protein nomenclature in the discussion part with regard to the cited animal species
- All gens and protein symbols should be styled according to species:
Eg. Humans, non-human primates and chicken: Full name: peroxisome proliferator–activated receptor γ
Gene symbol: PPARG (italic)
Protein symbol: PPARγ
Mice and rats: Full name: peroxisome proliferator–activated receptor γ
Gene symbol: Pparg (italic)
Protein symbol: PPARγ
Author Response
The manuscript entitled „Identification of novel insulin resistance related ceRNA network: LncRNA, RP11‐773H22.4; miR-3163, miR-1; mRNAs:RET , IGF1-R, m-TOR, GLUT-4, AKT2 in T2DM and its potential editing by CRISPR/Cas9” describe a innovative research aimed at constructing in siilco, a competing endogenous RNAs (ceRNAs) network linked to the pathogenesis of insulin resistance followed by its experimental validation in patients evaluation the efficacy of CRISPR/Cas9 as a potential therapeutic strategy to modulate the expression of this deregulated network in type 2 diabetic patients. The research was designed appropriately. The authors also describe all its phases in a very detailed way. Moreover the manuscript is very interesting and well written. Therefore, I have no major comments on its current form. However, please follow the manuscript from an editorial perspective as there are numerous spelling errors like e.g.
- invitro” without space.
As suggested by the reviewer we corrected it.
- two commas between authors.
As suggested by the reviewer we corrected it.
- please check also gene and protein nomenclature in the discussion part with regard to the cited animal species
We checked the nomenclature, and they are correct in relation with human species as we have no animal species names for genes in this article.
Reviewer 2 Report
In the manuscript “Identification of novel insulin resistance related ceRNA network: LncRNA, RP11‐773H22.4; miR-3163, miR-1; mRNAs:RET , IGF1-R, m-TOR, GLUT-4, AKT2 in T2DM and its potential editing by CRISPR/Cas9”, the authors proposed to unveil an endogenous RNAs (ceRNAs) network linked to the pathogenesis of insulin resistance and performed some validation experiments in patients’ and cell line samples. They also evaluated the efficacy of CRISPR/Cas9 as a potential therapeutic strategy to modulate the expression of this dysregulated network. The paper has a very interesting approach and could be of interest but there are too many drawbacks that hampered the initial enthusiasm. Although the approach may be valid, it stands as merely descriptive without a clear mechanistic approach. All sections need extensive revision, including the title. It is too long and very difficult to assess the interest on the paper. Figures are very amateur and need extensive revision. Overall the paper is technically interesting and could be of interest but all sections need extensive revision.
Specific comments:
- Language revision is mandatory. There are too many typo and grammar errors. For instance, in graphical abstracts “Bioinformatics” is written as “Bioiformatics”. Please carefully read the document and correct the errors.
- Methods must be revised. There are some crucial information missing. For instance, antibodies catalog number and dilutions should be included.
- Statistics are very incomplete. How did the authors reached the number of patients for validation? Power analysis? In addition, the impact of age on data analysis should have been better cautioned.
- Data presentation is not convincing. For instance, table 2 should be presented as a graph. Figure 1 is also not convincing as the data should be presented towards the control. That group should be on the left. Figure 1a and b should be merged as one. Data presentation is very amateur.
- Some subtitles are also very amateur. For instance: “Effect of CRISPR/Cas9 editing on GLUT-4 and m-TOR proteins (major effectors in the insulin signaling pathway)”. Why include the information that they are effectors in insulin signalling? Paper organization needs extensive revision.
- Discussion is disconnected and needs extensive revision to construct a coherent merge of the data with the current knowledge. The conceptual advance of the work is not on spotlight nor is well discussed. The discussion appears like disconnected subjects in a descriptive manner. There is no take home message nor any discussion on the clinical/biological significance of the data.
Author Response
In the manuscript “Identification of novel insulin resistance related ceRNA network: LncRNA, RP11‐773H22.4; miR-3163, miR-1; mRNAs:RET , IGF1-R, m-TOR, GLUT-4, AKT2 in T2DM and its potential editing by CRISPR/Cas9”, the authors proposed to unveil an endogenous RNAs (ceRNAs) network linked to the pathogenesis of insulin resistance and performed some validation experiments in patients’ and cell line samples. They also evaluated the efficacy of CRISPR/Cas9 as a potential therapeutic strategy to modulate the expression of this dysregulated network. The paper has a very interesting approach and could be of interest but there are too many drawbacks that hampered the initial enthusiasm. Although the approach may be valid, it stands as merely descriptive without a clear mechanistic approach. All sections need extensive revision, including the title. It is too long and very difficult to assess the interest on the paper.
As suggested by the reviewer we modified the tittle.
Figures are very amateur and need extensive revision. Overall, the paper is technically interesting and could be of interest, but all sections need extensive revision.
Specific comments:
- Language revision is mandatory. There are too many typo and grammar errors. For instance, in graphical abstracts “Bioinformatics” is written as “Bioiformatics”. Please carefully read the document and correct the errors.
As suggested by the reviewer we corrected it, and we revised the whole article for other typo and grammar errors.
- Methods must be revised. There are some crucial information missing. For instance, antibodies catalog number and dilutions should be included.
As suggested by the reviewer we completed all the missing cat no. and dilutions in methods.
- Statistics are very incomplete. How did the authors reach the number of patients for validation? Power analysis? In addition, the impact of age on data analysis should have been better cautioned.
Our study is a case control study. So, we usually calculate the values of Expected proportion exposed in controls and assumed odds ratio using the calculation found in this link http://epitools.ausvet.com.au/content.php?page=case-controlSS. The values of Expected proportion exposed in controls and assumed odds ratio were obtained from literatures, matching our study.
- Data presentation is not convincing. For instance, table 2 should be presented as a graph. Figure 1 is also not convincing as the data should be presented towards the control. That group should be on the left. Figure 1a and b should be merged as one. Data presentation is very amateur.
We agreed with reviewer that table 2 should be represented as graphs only. We deleted this table, and we present it as one merged boxplot graph in figure 1. Also,
As recommended by the reviewer, we modified figure 1 so as the control group is presented on left.
- Some subtitles are also very amateur. For instance: “Effect of CRISPR/Cas9 editing on GLUT-4 and m-TOR proteins (major effectors in the insulin signaling pathway)”. Why include the information that they are effectors in insulin signaling? Paper organization needs extensive revision.
As suggested by the reviewer we corrected the subtitle.
- Discussion is disconnected and needs extensive revision to construct a coherent merge of the data with the current knowledge. The conceptual advance of the work is not on spotlight nor is well discussed. The discussion appears like disconnected subjects in a descriptive manner. There is no take home message nor any discussion on the clinical/biological significance of the data.
As suggested by the reviewer we rewrote the discussion.
Reviewer 3 Report
This study has verified the genomic alignments related to insulin resistance in type 2 diabetes mellitus (T2DM) patients. Especially, the authors have focused on a long non-coding RNA, LncRNA-RP11-773H22.4, as a competing endogenous RNA. They are attempting a development of therapeutic application in insulin resistance by using CRISPR/Cas9 genome editing system for editing this alignment. It is interesting. However, there remain some points which should remedied in this paper. They are;
- In Discussion, the second paragraph. Those description are redundant. That should be remedied because there are same descriptions in Introduction.
- In Discussion, descriptions regarding cancer should be deleted because the readers may be confused. This article shows the insights about insulin resistance or T2DM.
- In Discussion P43, L5 – P44, L2, descriptions regarding miRNA. The influence of miRNA on T2DM is not clear in this study. The reviewer considered that behaviors of those miRNAs could depend on the lncRNA as well as other factors such as GLUT4 and mTOR etc. Therefore, the reviewer did not think that the miRNAs could directly influence the genes related to insulin resistance.
- Tables are so complicated. It is too tough to understand them.
Author Response
This study has verified the genomic alignments related to insulin resistance in type 2 diabetes mellitus (T2DM) patients. Especially, the authors have focused on a long non-coding RNA, LncRNA-RP11-773H22.4, as a competing endogenous RNA. They are attempting a development of therapeutic application in insulin resistance by using CRISPR/Cas9 genome editing system for editing this alignment. It is interesting. However, there remain some points which should remedied in this paper. They are;
- In Discussion, the second paragraph. Those description are redundant. That should be remedied because there are same descriptions in Introduction.
As suggested by the reviewer we remedied it.
- In Discussion, descriptions regarding cancer should be deleted because the readers may be confused. This article shows the insights about insulin resistance or T2DM.
As suggested by the reviewer we deleted it.
- In Discussion P43, L5 – P44, L2, descriptions regarding miRNA. The influence of miRNA on T2DM is not clear in this study. The reviewer considered that behaviors of those miRNAs could depend on the lncRNA as well as other factors such as GLUT4 and mTOR etc. Therefore, the reviewer did not think that the miRNAs could directly influence the genes related to insulin resistance.
I think we talked about miRNAs from all points of view as regard being important members in the ceRNA network and as crucial players in insulin resistance pathways. This has been clarified in the introduction.
- Tables are so complicated. It is too tough to understand them.
As suggested by the reviewer Tables are simplified by removing redundant data. Table 1 was simplified. We also deleted table 2 to be represented as a simplified boxplot figure1.
Old tables: Table 3, Table 4, Table 5 are now tables 2 ,3 ,4 and they are simplified.
Round 2
Reviewer 2 Report
The revised version of the manuscript has improved.
Reviewer 3 Report
The reviewer appreciates the response and effort which has been done by the authors to enhance the quality of the work. After the check and correction of misspelling is done, the manuscript would be acceptable.